# Controlled enzymatic synthesis of oligonucleotides
Maëva Pichon ⓘ & Marcel Hollenstein ⓘ ✉

Oligonucleotides are advancing as essential materials for the development of new therapeutics, artificial genes, or in storage of information applications. Hitherto, our capacity to write (i.e., synthesize) oligonucleotides is not as efficient as that to read (i.e., sequencing) DNA/RNA. Alternative, biocatalytic methods for the de novo synthesis of natural or modified oligonucleotides are in dire need to circumvent the limitations of traditional synthetic approaches. This Perspective article summarizes recent progress made in controlled enzymatic synthesis, where temporary blocked nucleotides are incorporated into immobilized primers by polymerases. While robust protocols have been established for DNA, RNA or XNA synthesis is more challenging. Nevertheless, using a suitable combination of protected nucleotides and polymerase has shown promises to produce RNA oligonucleotides even though the production of long DNA/RNA/XNA sequences (>1000 nt) remains challenging. We surmise that merging ligase- and polymerase-based synthesis would help to circumvent the current shortcomings of controlled enzymatic synthesis.

Over half a century ago, the structures as well as the biological functions of DNA and RNA have been unraveled. Since then, nucleic acids have advanced as key elements in numerous applications that markedly deviate from their natural functions. For instance, oligonucleotides are important building blocks in DNA-encoded libraries (DELs) for drug discovery campaigns[1,2], storage of digital information[3,4], and computing[5]. In addition, DNA and RNA can serve as potent catalysts[6–10], binders[11–17], and therapeutic agents[18,19]. The increasing popularity of nucleic acids is accompanied by an important surge in the demand of synthetic oligonucleotides, particularly of sequences containing chemical modifications.

Currently, two main strategies exist for the production of DNA and RNA, namely chemical, automated synthesis and polymerase-based approaches (Fig. 1). Automated synthesis using solid-phase phosphoramidite chemistry was first developed by Marvin Caruthers and Serge Beaucage in 1981[20,21]. The activated building blocks used in this approach consist of 5'-*O*-DMTr protected nucleosides equipped with reactive phosphoramidite moieties located at the 3' position of the sugar moiety. In an iterative process, phosphoramidite nucleosides are added on a growing chain which is immobilized on a solid support through a succession of cycles consisting of coupling, capping, oxidation, and deprotection steps. An interesting feat of this method is that it is directly amenable to the production of chemically modified oligonucleotides. The integration of chemical modifications is performed by substituting natural with modified phosphoramidites during solid phase synthesis. This technology is particularly useful for the crafting of therapeutic oligonucleotides (e.g., siRNAs

and antisense oligonucleotides (ASOs)) that bear multiple modification patterns as well as for xeno nucleic acids (XNAs) which consist of sometimes heavily modified sugar residues (*vide infra*). Due to these interesting properties, solid phase synthesis has advanced as the workhorse for the preparation of therapeutic oligonucleotides in industrial settings, particularly for sequences containing phosphate and sugar alterations[19,22].

Alternatively, oligonucleotides can be accessed via polymerase-mediated synthesis. These enzymes catalyze the co-polymerization of nucleoside triphosphates on DNA and RNA primers and produce sequences of variable lengths ranging from a few dozens to several thousands of nucleotides. This reaction can proceed either in a template-dependent or template-independent manner depending on the nature of the polymerase. Production of modified oligonucleotides is readily accessible by this approach provided the corresponding, chemically altered nucleoside triphosphates that are used *in lieu* of their natural counterparts are tolerated as substrates by the polymerases[23]. The potential of this method is showcased by the industrial production of mRNA vaccines which consist of long (>4000 nt), single-stranded RNA oligonucleotides containing modified nucleotides (mainly *N*1-methyl-pseudouridine)[24,25].

While both production methods of (modified) oligonucleotides are well-established, they suffer from shortcomings. Indeed, chemical synthesis based on the assembly of activated phosphoramidite building blocks is limited in terms of both scalability (<10 kg batches) and sustainability (over 4000 kg waste organic solvents and reagents per kg for a 20 nt long oligonucleotide[26–28]). In addition, even though coupling yields of

Institut Pasteur, Université Paris Cité, CNRS UMR3523, Department of Structural Biology and Chemistry, Laboratory for Bioorganic Chemistry of Nucleic Acids, 28, Rue du Docteur Roux, 75724 Paris Cedex 15, France. ✉e-mail: marcel.hollenstein@pasteur.fr

phosphoramidites are generally very high (≥99%), access to sequences larger than 150 nucleotides is difficult or impossible by this approach. This is due to the fact that the theoretical yields for larger sequences are low (e.g., 22% ($0.99^{150} \cdot 100\%$) for a 150-mer) and practical, isolated yields are much lower than these upper limits due to decreasing coupling efficiencies with increasing oligonucleotide lengths[29,30]. In addition, the rather harsh conditions (particularly during detritylation) imposed by solid-phase synthesis combined with errors in acetylation lead to depurination as well as single-base deletions or insertions (indels). The occurrence of such errors also increases with the length of the sequence and can have dire consequences. Even with an accuracy of 99.5% for adding a correct nucleotide, the overall fidelity for the production of a 150 nt long sequence would only be of 47% ($0.995^{150}$)[31]. The rather harsh conditions imposed by the synthetic cycles also restrict the choice of chemical modifications that can be introduced into oligonucleotides and is for instance not compatible with modifications that are susceptible to oxidation or react with electrophiles[32].

Similarly, while polymerase-mediated synthesis grants access to longer sequences (>1 kb), not all chemically modified nucleotides are tolerated by polymerases and specific introduction of single or multiple chemically altered species at discreet, user-defined locations is not possible. In addition, enantioselective synthesis with polymerases is still restricted to producing all-*Rp*[33-35] phosphorothioate linkages and the production of the required nucleoside triphosphates is labor intensive and low yielding[32,36,37].

In order to meet the exponential increase in demand of long and chemically modified oligonucleotides and to match the writing and reading speeds for storage of digital information applications, alternative, highly efficient, sustainable, and cost-affordable biocatalytic strategies are in dire need[38-40].

This Perspective provides an overview of existing and emerging biocatalytic methods for the production of oligonucleotides with a particular emphasis on controlled enzymatic synthesis. We also provide a description of other enzymatic methods and comment on the evolution and future directions of the field.

## Main
### Controlled enzymatic synthesis
Controlled enzymatic synthesis is a method for the de novo production of oligonucleotides that combines elements of solid-phase synthesis and of enzymatic polymerization of nucleotides (Fig. 1). Conceptually, temporarily blocked nucleotides, either at the level of the nucleobase or the 3'-position of the sugar moiety, are incorporated mainly by template-independent polymerases onto a solid support bound primer (Fig. 2). After removal of excess polymerase and triphosphate, the masking groups are removed, and the extended primer can be subjected to subsequent synthetic cycles. These transiently masking groups are referred to as clippable, because they need selective and efficient covalent bond breaking reactions to be removed[41].

While seemingly straightforward, this method requires extensive adjustments of the chemistry used for transitory protection of the nucleotide, the substrate tolerance of polymerases, and automation of the process. The design of the 3'-*O*-capping group is essential in controlled enzymatic synthesis since it will dictate the efficiency of the reaction, the viability of the process over numerous synthetic cycles, and the storage capacity of the modified nucleotide. Indeed, the transient blocking groups need to be robust enough to resist polymerase-mediated hydrolytic degradation[42-45] and support longer periods of storage in buffered aqueous solutions. Concomitantly, the masking group should be labile enough to permit fast and high yielding deprotection reactions under mild conditions that do not cause any hydrolysis of phosphodiester linkages nor affect the nucleotidic scaffold of constituting nucleotides of the sequence to be produced. Besides these considerations, the modified nucleotides should not only be tolerated by polymerases but act as excellent substrates so as to enable fast (i.e., within minutes) coupling times, exclusive formation of single addition products (which avoids the need for capping steps), and quantitative yields (essential to reach longer sequences). The masking group cannot be too bulky so as to avoid steric clashes with amino acid side chains of the active sites of the polymerase, which would preclude its successful incorporation into primers[46]. Lastly, in order to escape synthesis in the liquid bulk phase which only permits low-scale production of short sequences, nucleotides and polymerases both need to be compatible with solid support and multiplex approaches[47].

In the following section, we will describe progress made to produce natural (DNA and RNA) as well as chemically modified (XNAs) oligonucleotides with the pros and cons of each approach.

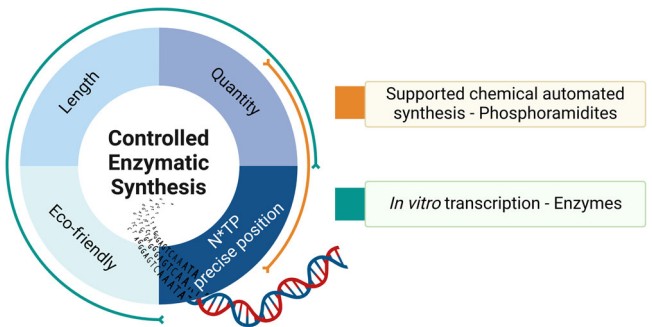

**Fig. 1 | Pros and cons of chemical automated synthesis and enzymatic synthesis of oligonucleotides.** Controlled enzymatic synthesis really resides at the interface of chemical and enzymatic approaches. Allowing theoretically to reach high quantity, high length and the use of modified nucleotides at the same time.

### Template-independent synthesis of DNA oligonucleotides
The development of methods for de novo enzymatic DNA synthesis have benefited from progress made in DNA sequencing by synthesis (SBS), an essential element in Sanger and Illumina sequencing methods. In SBS, an

**Fig. 2 | Schematic representation of de novo enzymatic synthesis of nucleic acids with template-independent polymerases.** Nucleoside triphosphates blocked at the level of the nucleobase or the sugar (mainly 3'-OH) are incorporated into single-stranded primers by polymerases. The transient blocking group prevents multiple incorporation events from occurring. Following polymerase-mediated incorporation, the masking group is removed and the system is ready for subsequent synthetic cycles.

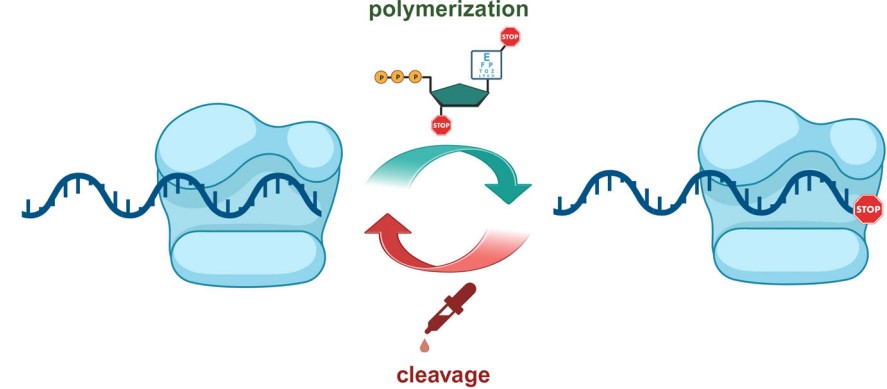

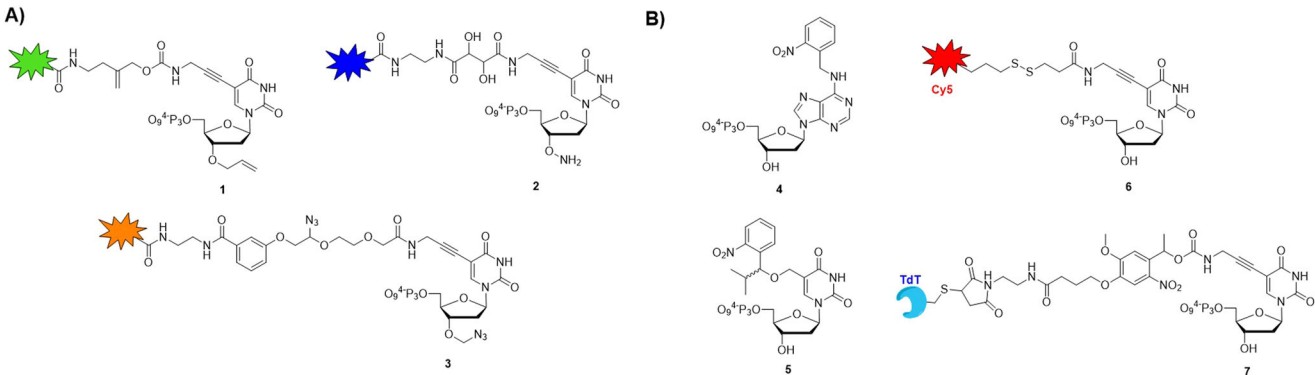

**Fig. 3 | Chemical structures of blocked nucleotides used in DNA sequencing by synthesis (SBS) and de novo DNA synthesis. A** Structures of common 3'-O-blocked nucleotides and (**B**) base-modified analogs. The colors represent different fluorophores or dyes that can be incorporated into DNA during SBS.

immobilized primer is extended by a single, fluorescently-labeled nucleotide, the nature of which is dictated by the templating nucleotide. The identity of the templating nucleotide is then identified by reading out the color of the dye, specific to each individual nucleotide. Sequencing then proceeds by unmasking the blocking group and repeating this cycle after a washing step[48–50]. Each of the nucleotides used in SBS carries a fluorophore that is connected to the nucleobase via a cleavable linker and is also equipped with a 3'-O-capping group that stops the enzymatic reaction after a single incorporation event[51–55]. Both modifications are removed simultaneously after the read-out step so as to restore a reactive 3'-OH moiety for subsequent SBS cycles and permit a correct identification of the next fluorescently-labeled nucleotide[56].

This impressive and important bulk of work allowed the identification of several capping moieties compatible with DNA polymerase synthesis and requiring only mild deprotection conditions. Amongst successful transient blocking moieties are the 3'-O-allyl group (nucleotide **1** in Fig. 3A) which are compatible with engineered polymerases such as the 9°N mutant DNA polymerase and can easily be removed in less than one minute by an in situ generated $Pd^0$ catalyst[57–60]. The smallest blocking group (besides methyl) that can be affixed at position 3' of a nucleotide is hydroxylamine ($O-NH_2$)[52]. The presence of such a small alteration on a nucleotide (structure **2** in Fig. 3A) is readily tolerated by a number of DNA polymerases (*vide infra*) including engineered Taq variants but also common Family A (e.g., *Bst* and Klenow) and Family B (e.g., $9°N_m$ and Therminator) DNA polymerases[61]. Besides displaying a high degree of compatibility with polymerases, this minimal capping moiety can also be removed by application of a mild oxidant ($NaNO_2$) that does not induce any significant oxidative damage to DNA[61]. Finally, the 3'-O-azidomethyl group is also capable of acting as a very potent reversible terminator (nucleotide **3** in Fig. 3A). Indeed, equipping nucleotides with such a capping moiety does not abrogate their capacity at serving as excellent substrates for the $9°N_m$ DNA polymerase and mutants thereof[54,55]. Azidomethyl groups can easily be removed by the action of mild reducing agents such as Tris(2-carboxyethyl) phosphine (TCEP) which first converts the azide to an amine via a Staudinger reaction followed by a water-mediated hydrolysis step[55]. Nucleotides equipped with fluorophores tethered to the nucleobases and reversibly terminated by azidomethyl groups have been commercialized by Illumina for sequencing applications. Other potentially useful, transient 3'-O-masking groups have also been proposed for such applications including carbonates[56], functionalized, photocleavable ethers[62], 3'-O-methyl[57], and 3'-O-2-cyanoethyl[51] which might be revisited in the future in the context of de novo enzymatic synthesis.

In addition to 3'-O-reversible terminators, modification of nucleobases at specific locations can also restrict DNA synthesis to single nucleotide insertions[63,64]. Initially, alkylation of 2'-deoxyadenosine with 2-nitrobenzyl bromide was believed to occur at the 3'-OH position of the nucleoside[57] (thus resulting in the formation of a 3'-reversible terminator), but a subsequent study confirmed alkylation at position N6 of the nucleobase rather

than at 3'-OH[65]. The $N^6$-alkylated dATPs (nucleotide **4** in Fig. 3) were then shown to be excellent substrates for commercial, template-dependent DNA polymerases such as *Bst* and result in single incorporation events during primer extension reactions with high mismatch discrimination capacity. As for 3'-O-masking groups, removal of the base-modification (e.g., by UV light irradiation) then permits enzymatic synthesis to resume. Nucleobase modification was then extended to pyrimidines, particularly to 5-substituted-aryloxy-methyl-dUTP analogs[66] and 5-propargyl-amino-dUTP scaffolds[67] (nucleotides **5** and **6** in Fig. 3, respectively). The presence of a rather bulky moiety inhibits the polymerase and prevents further incorporation events from occurring. While the exact mechanism has not yet been elucidated, fine tuning of the size of the nucleobase masking groups permits a modulation of both termination and the incorporation selectivity of the incoming nucleotide[66].

Collectively, development of robust SBS methods allowed to identify potent reversible terminators by modifying the 3'-O-position as well the nucleobase of nucleotides. These reversible terminators are fully compatible with enzymatic DNA synthesis, display high incorporation fidelities, lead to single incorporation events, and can easily be removed without inducing any damage to the extended DNA sequences. All these favorable features are directly exploitable and amenable to controlled enzymatic DNA synthesis. Nonetheless, fine-tuning and further engineering is still required to translate these findings to efficient and robust de novo DNA synthesis methods. Indeed, major limitations include the nature of the polymerases since SBS involves template-dependent polymerases (while de novo synthesis should preferentially rely on template-independent enzymes) and the nature of the blocking groups since nucleobase modification often leave a permanent molecular scar after deprotection (which can alter the properties of resulting oligonucleotides).

Based on these considerations, efforts at exploring controlled enzymatic DNA synthesis as a valid alternative to chemistry-based approaches for the production of oligonucleotides rapidly crystallized on the terminal deoxynucleotidyl transferase (TdT)[68–70]. The TdT is a member of the X family of DNA polymerases and catalyzes the addition of nucleotides at the 3'-termini of single-stranded DNA strands. In addition to being a template-independent polymerase, the TdT has also shown a rather high propensity at accepting modified nucleotides even when equipped with nucleobase surrogates[71,72] or sugar-modified entities[73–77]. The TdT can also extend oligonucleotides as short as three nucleotides in length and of different sequence composition and chemistries[78]. These favorable assets were already recognized in early reports which suggested the possibility of using blocked nucleotides to synthesize DNA[79] and RNA[80] oligonucleotides with the TdT as polymerase. Since then, various approaches have been reported. For instance, temporary blocking the 3'-OH of dNTPs with hydroxylamine was shown to be compatible with wild-type and even more so with mutant TdT variants[81], affording high yielding incorporation events (>99.5%) with all nucleotides[82]. The combination of 3'-O-$NH_2$-dNTPs and TdT variants has permitted the spatial microscale enzymatic synthesis of DNA with

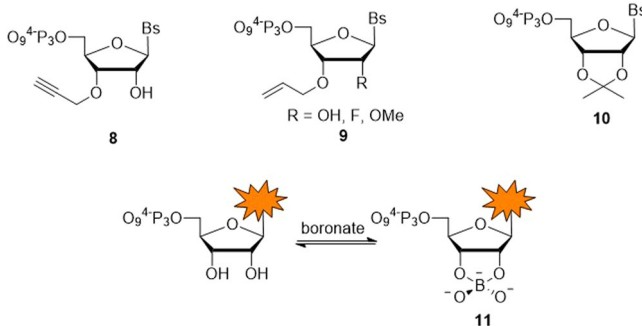

**Fig. 4 | Chemical structures of blocked RNA nucleotides.** The protection pattern can involve only 3'-*O*-protection (structures **8** and **9**) or *cis*-diol protection (nucleotides **10** and **11**). Bs represents any nucleobase.

single-base control. In this refined system, a precise silicon microelectromechanical system inkjet dispenses picoliters of the TdT enzyme alongside 3'-*O*-blocked nucleotides to perform a unique incorporation at the 3'-end of the DNA primer. The protecting group can then be cleaved and excess reagents removed by a washing step, liberating the 3'-OH position for further elongation steps[47]. This combination has also permitted the company DNA Script to launch the commercialization of the first, benchtop DNA synthesizer based on enzymatic incorporation of blocked nucleotides[39].

Similarly, 3'-*O*-azidomethyl-blocked nucleotides combined with the TdT are currently considered by the company Nuclera Nucleics to produce long DNA oligonucleotides. These blocked nucleotides were also used as substrates for other, template-dependent DNA polymerases in a cyclic reversible termination method. In this approach, oligonucleotides are immobilized on a solid support and designed so as to transiently form hairpins or duplexes with neighboring sequences. The polymerases are then capable of extending the 3'-termini of these transient structures with 3'-*O*-azidomethyl-based reversible terminators with yields exceeding 98%[83]. Other blocking groups that have been considered for de novo DNA synthesis in conjunction with the TdT include photocleavable, aryl ethers[84,85] and 3'-*O*-allyl[86].

Instead of using reversible terminators with 3'-*O*-masking groups, the TdT can be kinetically controlled by adding a competing enzyme. Particularly, the enzyme apyrase hydrolyzes nucleoside triphosphates into the corresponding di- and mono-phosphates and thus, depletes the pool of available TdT substrates in a controlled manner, in terms of number and sequence of nucleotide transitions[69]. While directly amenable to storage of digital information in DNA applications, this strategy could potentially be combined with blocked or partially modified nucleotides for de novo DNA synthesis.

Progress to further improve synthetic yields and reach yet longer sequences by this approach, will certainly involve exploring other template-independent polymerases such as polymerase theta[87]. In addition, numerous engineered TdT variants have already been screened and tested but their sequences and properties remain unknown due to intellectual property rights. Undoubtedly, the identification of yet more robust[88] and proficient TdT mutants[81] will also improve the effectiveness of controlled enzymatic DNA synthesis.

As for SBS methods, nucleobase-modified nucleotides can be used instead of 3'-*O*-reversible terminators. A prime example, pioneered by ANSA Biotechnologies, consists of TdT enzymes directly tethered to the nucleobase of a nucleotide via a linker arm (nucleotide **7** in Fig. 3). Directly connecting the TdT to the nucleobase permits the incorporation of a single, specific nucleotide into DNA and concomitantly accelerates the incorporation rate[89]. The exact mechanism underlying single incorporation events is still unclear but could stem from multiple origins. For instance, it is possible that the steric bulk caused by the presence of the tethered TdT prevents further addition from occurring. Also, rate acceleration (and hence

single incorporation) can be favored because the entropic penalty is less for an intra- vs. an inter-molecular reaction. Finally, the use of the TdT-nucleotide conjugate potentially increases the local concentration of both reaction partners and might coerce the active site of the TdT into a more suitable conformation. Some of these elements may also explain why other nucleobase-modified substrates such as **4** and **5** (Fig. 3) produce exclusively single incorporation events.

The insertion of photocleavable[89] or electrochemically[90] cleavable moieties between the TdT and the nucleobase then permits the removal of the polymerase under mild and efficient conditions. Average coupling times are below 2 min for C, G, and T and around 3 min for A, coupling yields of the TdT-modified nucleotides are in the 97–98% range, and deprotection is achieved by UV light irradiation within 1 min[89]. These highly advantageous features have recently been exploited for the production of a 1005-nucleotide long, unmodified DNA sequence[91]. Other base-blocked nucleotides, for instance photocaged deoxypseudouridine[92] or 5-hydroxymethyl-pyrimidines[93,94] have been reported recently but their compatibility with TdT-mediated enzymatic synthesis has not yet been evaluated.

Collectively, highly efficient approaches exist for controlled enzymatic DNA synthesis, either relying on 3'-*O*- or nucleobase-protection patterns, combined with variants of the template-independent DNA polymerase TdT. Undoubtedly, these robust technologies will be further extended for the production of very long (>1000 nt) or nucleobase-modified DNA oligonucleotides[95] in the near future.

## Template-independent synthesis of RNA oligonucleotides

Most research efforts for the development of controlled enzymatic synthesis have been devolved to DNA and de novo RNA synthesis by this approach remains an uncharted field of investigation. A structural key feature of RNA nucleotides is the presence of a *cis*-diol pattern where the 3'-OH is complemented by an additional hydroxyl moiety at position 2' of the sugar. Hence, the choice of the protection pattern in the design of reversible RNA terminators is not as straightforward as for DNA. Indeed, reversible terminators can consist of nucleotides either blocked at 3'-OH (and with a free 2'-OH) or at both positions (Fig. 4). Protection at a single 3'-OH position is sufficient once the nucleotide is installed on the primer since this group is directly involved in the $S_N2$ reaction with the α-phosphate moiety of the incoming nucleotide. However, such a single and selective protection pattern is synthetically more demanding and requires additional manipulation of nucleoside intermediates. Also, not all protecting groups are compatible with single protection patterns since they need to be resistant against 2',3'-migration during chemical and enzymatic synthesis and most protocols have been devised for the 2'-position rather than 3'-OH[96]. Despite more demanding synthetic campaigns, a single protection of the 3'-hydroxyl grants access to 2'-*O*-modified nucleotides which are prevalent in most therapeutic oligonucleotides[18]. These features have successfully been harnessed by the laboratory of Georges Church, who introduced 3'-*O*-propargyl- and 3'-*O*-allyl-masking groups on unmodified and 2'-*O*-modified ribonucleotides (structures **8** and **9** in Fig.4, respectively)[97]. These reverse terminators acted as excellent substrates for a mutant variant of the Poly(U) Polymerase (PUP). High coupling yields (>95%) could be achieved within a few minutes of primer extension reactions with all canonical and with 2'-*O*-modified nucleotides. Deblocking of the allyl moiety could be achieved in high yields in rather short periods of time (12 min). This method was successfully applied to multi-cycle enzymatic RNA synthesis, both in solution and on solid-support-bound RNA primers[97]. Up to 10 synthetic cycles could be achieved, which is a remarkable feature. Nonetheless, Pd-mediated deprotection steps are not compatible with phosphorothioate linkages and the presence of a free 2'-OH moiety is involved in the formation of 2'-5'-branched by-products, suggesting a small room for improvement of this approach.

On the other hand, a single, *cis*-2',3'-diol-protection step is synthetically much more affordable and also will prevent 2'-5'-branched RNA from forming during de novo RNA synthesis. However, this protection scheme abrogates the possibility of introducing therapeutically relevant, 2'-altered

**Fig. 5 | Chemical structures of XNA nucleotides.**
**A** Structures of 3'-*O*-blocked locked nucleic acids (LNAs) and **B** structures of threose nucleic acids (TNAs) and hexitol nucleic acids (HNAs) nucleotides. Bs represents any nucleobase.

nucleotides. Concomitantly, our laboratory has identified a number of potential *cis*-diol protecting groups for the design of RNA reversible terminators[98]. Pyrimidine nucleotides equipped with a 2',3'-O-iso-propylidene masking group (structure **10** in Fig. 4) acted as excellent substrates for wild-type PUP and afforded high coupling yields (>95%) in less than 5 min of primer extension reactions, but purines equipped with a similar blocking scaffold were not tolerated by wild-type PUP or the corresponding polyadenosine polymerase (PAP). A conceptually related approach was reported by Kyoung et al. who used the in situ formation of a transient ribonucleotide–borate complex (structure **11** in Fig. 4) to 3'-terminally label RNA oligonucleotides with the TdT[99].

Collectively, these recent reports clearly underscore the possibility of using RNA reversible terminators combined with the template-independent RNA polymerases PAP and PUP for de novo RNA synthesis. While the TdT has shown a limited yet non negligible degree of compatibility with modified ribonucleotides[68,99–101], PUP/PAP appear as more suitable enzymes for controlled enzymatic RNA synthesis. Masking scaffolds consisting of single, 3'-*O*-ether moieties or *cis*-diol protecting groups can both be considered for the construction of reversible RNA terminators. These recent reports will certainly serve as robust foundations for the construction of longer (100-200 nt) RNA oligonucleotides with or without modified residues in the near future.

**Extension to sugar-modified nucleic acids (XNAs)**
Xeno nucleic acids (XNAs) are a class of synthetic genetic polymers that entail a substantially different sugar composition compared to that of canonical DNA and RNA, and their replication is orthogonal to that of the natural counterparts (Fig. 5)[102,103]. Besides the creation of genetically modified organisms, XNAs are key elements in the development of potent therapeutic oligonucleotides (i.e., antisense therapy, aptamers, and XNA enzymes) due to their enhanced nuclease resistance[10,15,104] and in other applications such as mediation of cellular aggregation *via* their orthogonal base-pairing nature[105,106] or the construction of smart biomaterials[107]. Given the increasing popularity of XNAs, alternative methods to chemical synthesis and the use of engineered polymerases are in dire need.

Theoretically, controlled enzymatic XNA synthesis using reversible terminators should be feasible. However, XNAs face a number of additional challenges in comparison to DNA/RNA and nucleic acids with less imposing modification patterns. Indeed, more intricate and demanding synthetic routes are required for the production of suitably protected XNA nucleosides and nucleotides[18] and even alternate phosphorylation protocols might be necessary[108,109]. Moreover, given the important alterations in the chemical nature of the backbone, XNA nucleotides are often poor substrates for existing DNA and RNA polymerases.

Our laboratory has explored several transient 3'-*O*-masking groups for locked nucleic acids (LNA) nucleotides (structures **12a**-**12d** in Fig. 5A) and evaluated their capacity at serving as substrates for DNA polymerases for controlled enzymatic XNA synthesis[35,44,86,110]. The installation of a 3'-phosphate moiety on LNA-TTP (nucleotide **12a** in Fig. 5A) decreased the substrate tolerance of engineered, template-dependent DNA polymerases as

well as the TdT. Nonetheless, the major hurdle caused by the presence of an additional phosphate moiety, even on canonical DNA nucleotides, was the inherent phosphatase activity of a number of DNA polymerases, suggesting that such a transient protecting group was not compatible with de novo nucleic acid synthesis[44]. In order to act against the esterase/phosphatase activity of polymerases, more robust esters (nucleotides **12b** and **12c** in Fig. 5A) as well as ethers (nucleotide **12d** in Fig. 5A) were introduced into LNA nucleotides[86,110]. These sturdy groups successfully resisted against polymerase-mediated hydrolysis and were compatible with template-dependent DNA polymerases. Primer extension reactions revealed that mainly single LNA incorporation events could be achieved, albeit after rather long (>5 h) incubation times. Interestingly, even though both the TdT and the PUP readily accepted a number of 3'-*O*-blocked LNA nucleotides as substrates[86,110], both polymerases stalled after the introduction of a single, protected or unprotected, LNA nucleotide[75]. Threose nucleic acids (TNAs) nucleotides (structure **13** in Fig. 5B) have recently been evaluated as substrates for the TdT[107]. While being better substrates than LNA nucleotides, only a few TNA nucleotides can be incorporated at the 3'-termini of DNA primers by the TdT, even after several hours of reaction. Unlike other XNAs, hexitol nucleic acids (HNA, structure **14** in Fig. 5B) are rather well-tolerated as substrates by the TdT, even better than ribonucleotides, since up to 15 purine HNA nucleotides and four pyrimidine HNA nucleotides can be incorporated into DNA primers[111]. Nonetheless, these studies suggest that engineered TdT and PUP polymerases will be required for de novo XNA synthesis applications.

**Template-dependent enzymatic approaches for nucleic acid synthesis**
In parallel to controlled enzymatic synthesis of oligonucleotides, various biocatalytic approaches based on template-dependent polymerases have been reported. For instance, asymmetric PCR[112], Nicking Enzyme Amplification Reaction[113,114], and primer extension reactions followed by magnoseparation[115] or digestion[116] have been proposed. More recently, an elegant approach for the crafting of short, modified therapeutic oligonucleotides is based on the polymerization of modified nucleotides on a catalytic self-priming hairpin template (Fig. 6)[117]. Briefly, the strategy developed by the Lovelock group is based on the use of a DNA polymerase (*Thermococcus kodakarensis* KOD1) with a rather large tolerance for modified nucleoside triphosphate building blocks for the extension of a self-priming hairpin template. The fully extended template is then divided into expected, modified product and initial template via the action of an endonuclease (*Thermotoga neapolitana* *Tn*EndoV) which cleaves the second phosphodiester moiety after a deoxyinosine moiety installed on the template. This cleavage reaction regenerates the hairpin template which can be used for another one-pot enzymatic cascade cycle. Advantages of this approach encompass a minimum amount of by-products, with only pyrophosphate expelled after primer extension reactions and compatibility with a large number of chemical modifications including phosphorothioates, 2'-fluoro-, 2'-OMe, and LNA. Application of this biocatalytic approach allowed to produce several clinically relevant oligonucleotides in

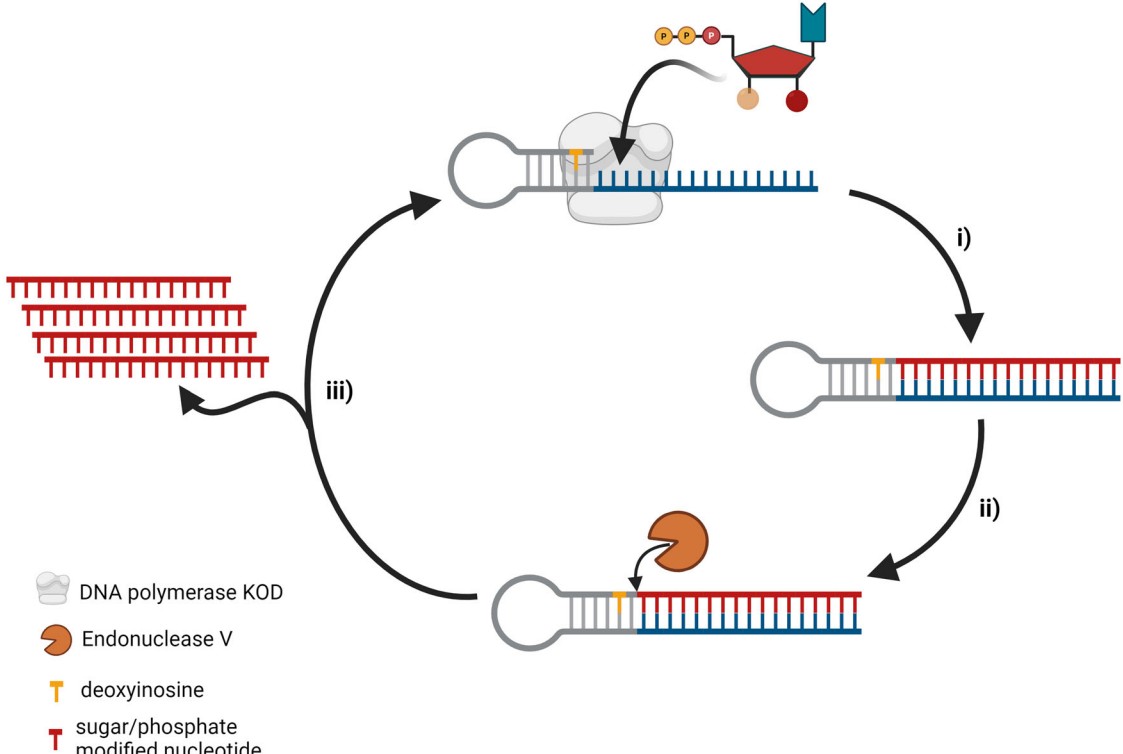

**Fig. 6 | Alternative biocatalytic method for the production of modified nucleic acids.** This isothermal biocatalytic approach based on the polymerase-mediated incorporation of modified nucleotides into a self-priming hairpin template (gray–blue). (i) Template-dependent synthesis using sugar and phosphate-modified nucleotides and the KOD polymerase; (ii) endonuclease V cleaves the second phosphodiester linkage 3'-downstream of a deoxyinosine moiety present in the template and generates the modified sequence (in red); (iii) after endonuclease-cleavage, the template is available for subsequent rounds of catalytic synthesis.

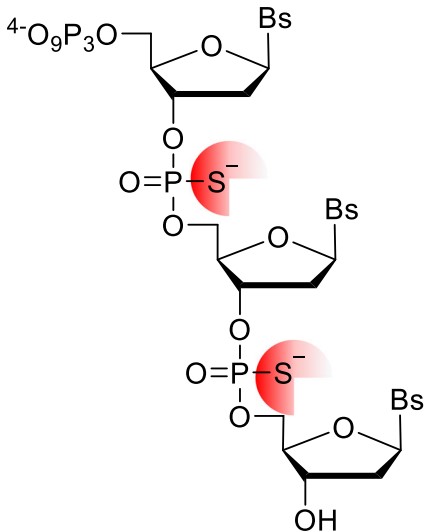

**Fig. 7 | Enzymatic synthesis of DNA three-by-three rather than one-by-one.** Chemical structure of a trinucleotide triphosphate stabilized by phosphorothioate linkages. Bs represents any nucleobase.

up to 100 mg scales. Finally, the process mass intensity (PMI) was estimated at 530 kg of raw material for 1 kg of oligonucleotide product compared to >4000 kg for phosphoramidite-based synthesis. Potentially, some of the 3'-O-blocked nucleotides described in the above sections could be compatible with this isothermal biocatalytic approach for the introduction of XNAs at site-specific positions in oligonucleotides.

Another, potentially useful approach relies on the use of a trinucleotide codon rather than a single nucleotide as substrates for polymerases (Fig. 7).

The replacement of internal phosphodiester bonds of trinucleotide triphosphates with phosphorothioates moieties permits to resist against the phosphatase activity of DNA polymerases and concomitantly supports enzymatic synthesis of DNA three-by-three[118]. While this method has only been evaluated on rather simple nucleotidic scaffolds so far, it could be extended to XNA-containing trinucleotides or codons capped with transient blocking groups at the 3'-termini for controlled enzymatic XNA synthesis.

In addition to these alternative, biocatalytic approaches, engineering of more potent template-independent polymerases, either by rational design[119–121] or evolution experiments[122,123], will certainly permit controlled enzymatic synthesis of RNA and XNA oligonucleotides in the near future.

### Ligase-based strategies

Despite impressive advancements in controlled enzymatic synthesis of nucleic acids, this approach still needs to overcome serious limitations. While robust protecting groups for reversible terminators have been identified for DNA, the situation is less clear for RNA, modified DNA/RNA, and particularly for XNAs. What is more, the absence of suitable protecting groups and especially of template-independent polymerases capable of recognizing XNAs and to a certain extent, modified RNAs, pushes controlled enzymatic synthesis of such biopolymers back into a relative stage of infancy. Finally, even though a >1 kb DNA fragment could be achieved by controlled enzymatic DNA synthesis, automation, larger scale production, and extension to yet larger oligonucleotides still remain important challenges. To by-pass a number of these hurdles, biocatalytic approaches relying on other enzymes than polymerases have been considered[40]. Amongst interesting enzymes are DNA and RNA ligases which catalyze, often via the help of ATP, the formation of covalent phosphodiester bonds between the 3'-OH of an acceptor strand and a 5'-phosphate moiety of a donor sequence. Surprisingly, DNA ligases are rather lax in terms of substrate requirements and tolerate the presence of modifications on the

nucleobase[124–126], but also on the sugar moiety of oligonucleotides[127]. This rather high permissiveness can be exploited to generate longer, chemically modified oligonucleotides by joining together carefully designed precursor fragments. This technology has been successful for sequences containing 2'-O-modifications but also XNAs such as HNA and TNA (Fig. 5B)[127,128]. As for polymerases, the substrate scope and catalytic promiscuity of ligases for modified systems can be improved by molecular evolution or rational design approaches[129]. For instance, the introduction of a single glycine residue at the hinge region at the interface between the oligonucleotide binding and the nucleotidyl-transferase domains massively improved the ligation capacity of the enzyme which could connect two, fully modified XNA (2'-OMe and HNA) oligonucleotides on an XNA template, a feature not accessible to any existing ligase[129]. Alternatively, our laboratory has shown that short (i.e., pentanucleotide) monophosphorylated fragments containing a variety of sugar, phosphate, and nucleobase modifications could be combined together using a suitably designed DNA template and the T3 DNA ligase[130]. This isothermal, one-step approach permits the production of short, clinically relevant oligonucleotides, but also much larger (>150 nt), heavily modified (>50 chemically modified nucleotides) fragments which are not readily accessible by any existing chemical or enzymatic approach. A related strategy was recently reported for the construction of short RNA sequences bearing 2'-O-modifications[131].

In the future, a combination of ligases and polymerases, as reported for the MEGAA method for DNA variant synthesis[132] or the Gibson assembly[133], might be leveraged to access long XNA oligonucleotides.

## Summary and outlook

Collectively, nucleic acids and their chemically modified counterparts have advanced as major players in numerous fields including therapeutics, diagnostics, storage technology, and nanotechnology. Given the importance of these biopolymers, the demand for oligonucleotides is exponentially increasing. Moreover, these needs increase not only in terms of number of sequences, but also with respect to length of oligonucleotides and presence and specific localization of an increasing variety of chemical modifications. Traditional chemical and enzymatic methods are overwhelmed by the sheer demand but are also impinged by technical limitations due to increasing complexity of the chemical nature of desired oligonucleotides. For instance, the production of a 150 nt RNA sequence equipped with dozens of individual modifications at user-defined, specific locations on a reasonable scale (>10 nmol) is a daunting task for any existing method. Consequently, alternative, biocatalytic approaches are in dire need. Amongst various approaches, controlled enzymatic synthesis is certainly the most advanced. In this chemoenzymatic nucleic acid synthesis method, transiently blocked nucleotides are incorporated by (mainly template-independent) polymerases and after installation of a single nucleotide and deprotection of the masking group, additional synthetic cycles are performed. This method is particularly well-developed for unmodified DNA since a first DNA synthesizer prototype has been commercialized recently and oligonucleotides of up to 1 kb are accessible. Nonetheless, challenges in the field still need to be lifted. Particularly, suitable protecting groups for RNA, and more importantly for modified nucleic acids such as XNAs still need to be identified. Additionally, more processive polymerases, particularly with XNA substrates need to be engineered for this method to be generally applicable. In parallel to controlled enzymatic synthesis, other polymerase-based approaches are currently evaluated and research interest shifts towards yet unrelated enzymes such as ligases.

With recent progress in enzyme engineering and design combined with medicinal chemistry type of approaches for the identification of potential nucleobase or sugar masking groups, controlled enzymatic synthesis has a bright future. Progress in the field will also make this approach more cost-affordable and eventually cheaper than existing, traditional methods. The potential of this method can be further enhanced by combining other enzymes such as ligases and potentially will enable the production of any type and size of oligonucleotide, modified or not, on industrial scales in the not-so-distant future.

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

## Acknowledgements

The authors acknowledge generous funding from DARRI and Institut Carnot "Pasteur Microbes and Health" Call 2021 (grant # INNOV-99-2, including a postdoctoral fellowship to M.P.). Institut Pasteur is acknowledged for funding.

## Author contributions

M.P. and M.H. wrote and edited the manuscript and M.H. conceived the scope and constructed the manuscript.

## Competing interests
The authors declare no competing interests.
