## [Peer Review File · Communications Chemistry]

Reviewers' comments:

Reviewer #1 (Remarks to the Author):

In this perspective review, Pichon and Hollenstein are discussing the recent advances made in synthesizing nucleic acids using enzymatic approaches, which have recently showed up as a plausible alternative to standard chemical solid-phase synthesis. Authors have chosen to organize the discussion based on the intended product: DNA, RNA or nucleic acid analogs, and then expand their presentation towards less "controlled" synthesis conditions with ligases or template-dependent polymerases. You can argue that there is still a level of control when manipulating conventional polymerases, or ligating fragments together, but it clearly departs from the focus on template-independent enzymes. However, given that this is a perspective article, I believe it is encouraged to write in such a way that it promotes scientific discussion and expanding to other enzyme families is within the scope of such article types. The information flows well, the bibliography is rich and cites relevant sources. In my opinion, the article is in publishable form, but I would like to draw the authors' attention to a couple of minor points:

The whole concept of controlled enzymatic synthesis seems to be supported by commendable goals, such as overcoming length limitation, sustainability, increasing the writing throughput, etc. At some point, the authors mention cost-affordable. Is it something that would be worth discussing, as part of the authors' opinion on the matter? Namely, do we have any way to compare the writing costs of DNA synthesis in the traditional sense to writing DNA using the TdT (to keep it simple). Surely we are in no way close to the costs of the current technology, but is there any available metric to gauge how much more expensive de novo synthesis still is? The technology is in its infancy and most players are from the industry, so this might be an apples and oranges situation, but I was wondering if there are estimates, or if it's shown clear cost improvements since the TdT has shown up in the field. Maybe the simplest would be to compare the benchtop synthesizer from DNAScript and the costs of consumables.

In a similar fashion, namely bringing the technology to being widely adopted by the scientific community, do we know anything regarding the TdT mutants? We seem to have a good idea of how to prepare triphosphates for controlled nucleotide addition, and we seem to believe that mutating TdT is going to be a major development towards speed and efficiency, but can anyone comment on how much they were mutated, or where, or how mutation has affected promiscuity?

Very minor comment:

In the intro, I would mention how depurination is also, eventually, responsible for a decrease in yield and quality

page 7, line 272-273: "because the entropic penalty is less for an intra- vs an inter-molecular reaction" regarding the use of TdT-dNTP conjugates. Wouldn't the accelerated rate of incorporation also be due to the increased local concentration of all relevant actors in the extension process? I am not saying there isn't any decreased entropic cost in carrying out an intramolecular reaction, but I wonder if the covalent attachment of an NTP to the enzyme isn't also forcing the enzyme to adopt an unusual conformation in order to properly present the NTP to the primer? Unless the tethered dNTP was engineered to fit perfectly snugly in the active site of the enzyme.

Reviewer #2 (Remarks to the Author):

This is a very nice review on a timely and important topic of enzymatic synthesis of oligonucleotides. As it only covers the synthesis of single-stranded ONs, it may be good to include the wording single-stranded in the title and abstract.

It covers two principally different approaches - template independent and template dependent - this subdivision should be in the subtitles of the review (rather than "controlled enzymatic synthesis" and "other polymerase based approaches...").

The template independent methods based on TdT are covered exhaustively and this part is very well written.

The part "Other..." - which should be called "Template-dependent..." is not comprehensive and there are several important methods completely missing:

1. synthesis of long modified ONs using asymmetric PCR - e.g. <https://doi.org/10.1093/nar/gkaa999>

2. synthesis of modified ssONs using PEX followed by magnetoseparation with SA-coated magnetic beads - e.g. <https://doi.org/10.1002/chem.200701249>

3. synthesis of modified ssONs using PEX with 5'-phosphorylated template followed by digestion of the template with lambda-exonuclease:

<https://doi.org/10.1007/BF02740823>

<https://doi.org/10.1039/A908022H>

<https://doi.org/10.1093/nar/gkaa999>

4. synthesis of shorter modified ONs using Nicking Enzyme Amplification Reaction (NEAR) -

<https://doi.org/10.1039/C2CC32930A>

<https://doi.org/10.1021/bc400149q>

5. synthesis of modified ODNs using reverse-transcription from RNA templates followed by digestion of the RNA - <https://doi.org/10.1039/D2CC03588J>

After incorporation of these methods and revising the corresponding discussion, the review will be perfectly acceptable for publication in CommChem

Reviewer #3 (Remarks to the Author):

Oligonucleotides are fundamentally important materials for therapeutics, biotechnology and information storage. In this manuscript the authors comprehensively summarized the recent advances of controlled enzymatic synthesis of oligonucleotides, including natural DNA and RNA as well as chemically modified XNA. The manuscript is clearly organized and well written, and will be a good read to the interested readers. I recommend publication in Communications Chemistry after a minor revision.

1. Formatting issues, grammar errors and typos should be corrected.

a) Page 2, line 75, "the potential of this is method is showcased by ...". The extra "is" should be deleted.

b) Page 4, line 154, "nucleobase of the sugar". The word "of" should be changed to "or".

c) Page 8, line 352, "aptamers and DNA enzymes". I believe in this section the authors meant to talk about XNA enzymes instead of DNA enzymes.

d) Page 10, line 428, "approach relies on the of a trinucleotide codon". I believe the word "use" is missing here.

2. Consistency. Page 9, figure 5. The triphosphate groups in panels a and b should be presented in a consistent format.

3. Page 9, the title of the last section. Since the following paragraph mainly describes a template-directed oligonucleotide synthesis, the word "de novo" in the title is not accurate.

4. In addition to TdT and PUP, other template-independent polymerases are also potentially useful for controlled enzymatic synthesis of oligonucleotides. For example, DNA polymerase theta mutants have been used for enzymatic synthesis of RNA (analogue) sequences (Nucleic Acids Res., 2018, 46, 6271), which should be cited.

Reviewer #1 (Remarks to the Author):

In this perspective review, Pichon and Hollenstein are discussing the recent advances made in synthesizing nucleic acids using enzymatic approaches, which have recently showed up as a plausible alternative to standard chemical solid-phase synthesis. Authors have chosen to organize the discussion based on the intended product: DNA, RNA or nucleic acid analogs, and then expand their presentation towards less "controlled" synthesis conditions with ligases or template-dependent polymerases. You can argue that there is still a level of control when manipulating conventional polymerases, or ligating fragments together, but it clearly departs from the focus on template-independent enzymes. However, given that this is a perspective article, I believe it is encouraged to write in such a way that it promotes scientific discussion and expanding to other enzyme families is within the scope of such article types. The information flows well, the bibliography is rich and cites relevant sources. In my opinion, the article is in publishable form, but I would like to draw the authors' attention to a couple of minor points:

Response: We thank the reviewer for the positive assessment of our manuscript.

The whole concept of controlled enzymatic synthesis seems to be supported by commendable goals, such as overcoming length limitation, sustainability, increasing the writing throughput, etc. At some point, the authors mention cost-affordable. Is it something that would be worth discussing, as part of the authors' opinion on the matter? Namely, do we have any way to compare the writing costs of DNA synthesis in the traditional sense to writing DNA using the TdT (to keep it simple). Surely we are in no way close to the costs of the current technology, but is there any available metric to gauge how much more expensive de novo synthesis still is? The technology is in its infancy and most players are from the industry, so this might be an apples and oranges situation, but I was wondering if there are estimates, or if it's shown clear cost improvements since the TdT has shown up in the field. Maybe the simplest would be to compare the benchtop synthesizer from DNAScript and the costs of consumables.

Response: We appreciate the reviewer's comment on the importance of cost evaluation. As the reviewer mentions, de novo enzymatic synthesis is in a relative stage of infancy and estimating the costs and comparing these to traditional solid-phase synthesis is rather difficult. In addition, DNA synthesis is much more developed than RNA or XNAs and therefore cost estimations might be even more arduous in the latter. Even though cost-evaluation is a bit premature and beyond the scope of the current Perspective article, we have now included the following sentence in the conclusion section:

Progress in the field will also make this approach more cost-affordable and eventually cheaper than existing, traditional methods.

In a similar fashion, namely bringing the technology to being widely adopted by the scientific community, do we know anything regarding the TdT mutants? We seem to have a good idea of how to prepare triphosphates for controlled nucleotide addition, and we seem to believe that mutating TdT is going to be a major development towards speed and efficiency, but can anyone comment on how much they were mutated, or where, or how mutation has affected promiscuity?

Response: We thank this reviewer for this important and interesting comment. As most TdT mutants used in the context of controlled enzymatic DNA synthesis have been developed by company they are covered by patents and hence very little information is available. In addition to the discussion of the very few articles that describe mutant TdT enzymes (reference #81 of the original manuscript) in the first version of this manuscript, we have now added the following sentence to clarify this point:

In addition, numerous engineered TdT variants have already been screened and tested but their sequences and properties remain unknown due to intellectual property rights. Undoubtedly, the identification of yet more robust⁸⁸ and proficient TdT mutants⁸¹ will also improve the effectiveness of controlled enzymatic DNA synthesis.

We have also inserted a citation to the following reference:

88 Chua, J. P. S. *et al.* Evolving a Thermostable Terminal Deoxynucleotidyl Transferase. *ACS Synth. Biol.* **9**, 1725-1735, doi:10.1021/acssynbio.0c00078 (2020).

Please also refer to our response to comment #4 by reviewer #3.

Very minor comment:

In the intro, I would mention how depurination is also, eventually, responsible for a decrease in yield and quality

Response: We thank the reviewer for this comment. We have now included this in the introduction section.

page 7, line 272-273: "because the entropic penalty is less for an intra- vs an inter-molecular reaction" regarding the use of TdT-dNTP conjugates. Wouldn't the accelerated rate of incorporation also be due to the increased local concentration of all relevant actors in the extension process? I am not saying there isn't any decreased entropic cost in carrying out an intramolecular reaction, but I wonder if the covalent attachment of an NTP to the enzyme isn't also forcing the enzyme to adopt an unusual conformation in order to properly present the NTP to the primer? Unless the tethered dNTP was engineered to fit perfectly snugly in the active site of the enzyme.

Response: We thank the reviewer for this very interesting comment/question. In the absence of mechanistic studies, this remains an unanswered question. We have also been wondering what the mechanism of the TdT-dNTP conjugates really is, but also that for the N6-alkylated dATPs and the 5-substituted-aryloxy-methyl-dUTPs mentioned earlier in our manuscript. It is possible that the reasons mentioned by this reviewer combined with the entropic penalty mentioned in the original manuscript might be responsible for all this. Hence, we have now changed the entire discussion to consider all these considerations to the following section:

The exact mechanism underlying single incorporation events is still unclear but could stem from multiple origins. For instance, it is possible that the steric bulk caused by the presence of the tethered TdT prevents further addition from occurring. Also, rate acceleration (and hence single incorporation) can be favored because the entropic penalty is less for an intra- vs an inter-molecular reaction. Finally, the use of the TdT-nucleotide conjugate potentially increases the local concentration of both reaction partners and might coerce the active site of the TdT into a more suitable conformation. Some of these elements may also explain why other

nucleobase-modified substrates such as **4** and **5** (Fig. 3) produce exclusively single incorporation events.

Reviewer #2 (Remarks to the Author):

This is a very nice review on a timely and important topic of enzymatic synthesis of oligonucleotides. As it only covers the synthesis of single-stranded ONs, it may be good to include the wording single-stranded in the title and abstract.

It covers two principally different approaches - template independent and template dependent - this subdivision should be in the subtitles of the review (rather than "controlled enzymatic synthesis" and "other polymerase based approaches...").

The template independent methods based on TdT are covered exhaustively and this part is very well written.

Response: We thank the reviewer for the positive assessment of our manuscript. We have now changed the subtitles of our manuscript accordingly.

The part "Other..." - which should be called "Template-dependent..." is not comprehensive and there are several important methods completely missing:

1. synthesis of long modified ONs using asymmetric PCR - e.g. <https://doi.org/10.1093/nar/gkaa999>

2. synthesis of modified ssONs using PEX followed by magnetoseparation with SA-coated magnetic beads - e.g. <https://doi.org/10.1002/chem.200701249>

3. synthesis of modified ssONs using PEX with 5'-phosphorylated template followed by digestion of the template with lambda-exonuclease:

<https://doi.org/10.1007/BF02740823>

<https://doi.org/10.1039/A908022H>

<https://doi.org/10.1093/nar/gkaa999>

4. synthesis of shorter modified ONs using Nicking Enzyme Amplification Reaction (NEAR) -

<https://doi.org/10.1039/C2CC32930A>

<https://doi.org/10.1021/bc400149q>

5. synthesis of modified ODNs using reverse-transcription from RNA templates followed by digestion of the RNA - <https://doi.org/10.1039/D2CC03588J>

After incorporation of these methods and revising the corresponding discussion, the review will be perfectly acceptable for publication in CommChem

Response: We thank the reviewer for this important comment. We fully agree that the methods mentioned by this reviewer were omitted in our original manuscript. The reason for this is that we were invited to contribute to the journal with a perspective and not a review article on the specific topic of controlled enzymatic synthesis. Hence, we were limited in terms of references, topics that can be covered, and length of the manuscript which should be rather concise. Nonetheless, since we totally agree with the reviewer that these are important methods for the generation of modified nucleic acids. We have now included a succinct summary of the methods mentioned in points #1, #2, #4, and #5 and cited the references as suggested. However, for the sake of size and limit of references, we have not considered λ -exonuclease digestion of modified dsDNA in our manuscript because this is a well-established and general method for the conversion of dsDNA/XNA to ssDNA/XNA rather for the direct, chemoenzymatic production of modified nucleotides. We have also changed the title of this section as suggested by the reviewer and we have included the following section in our manuscript:

For instance, asymmetric PCR¹¹², Nicking Enzyme Amplification Reaction^{113,114}, and primer extension reactions followed by magnetoseparation¹¹⁵ or digestion¹¹⁶ have been proposed.

The following references were included:

- 112 Ondruš, M., Sýkorová, V., Bednářová, L., Pohl, R. & Hocek, M. Enzymatic synthesis of hypermodified DNA polymers for sequence-specific display of four different hydrophobic groups. *Nucleic Acids Res.* **48**, 11982-11993, doi:10.1093/nar/gkaa999 (2020).
- 113 Měnová, P. & Hocek, M. Preparation of short cytosine-modified oligonucleotides by nicking enzyme amplification reaction. *Chem. Commun.* **48**, 6921-6923, doi:10.1039/C2CC32930A (2012).
- 114 Měnová, P., Raindlová, V. & Hocek, M. Scope and Limitations of the Nicking Enzyme Amplification Reaction for the Synthesis of Base-Modified Oligonucleotides and Primers for PCR. *Bioconjugate Chem.* **24**, 1081-1093, doi:10.1021/bc400149q (2013).
- 115 Brázdilová, P. *et al.* Ferrocenylethynyl Derivatives of Nucleoside Triphosphates: Synthesis, Incorporation, Electrochemistry, and Bioanalytical Applications. *Chem. Eur. J.* **13**, 9527-9533, doi:<https://doi.org/10.1002/chem.200701249> (2007).
- 116 Ondruš, M., Sýkorová, V. & Hocek, M. Traceless enzymatic synthesis of monodispersed hypermodified oligodeoxyribonucleotide polymers from RNA templates. *Chem. Commun.* **58**, 11248-11251, doi:10.1039/D2CC03588J (2022).

Reviewer #3 (Remarks to the Author):

Oligonucleotides are fundamentally important materials for therapeutics, biotechnology and information storage. In this manuscript the authors comprehensively summarized the recent advances of controlled enzymatic synthesis of oligonucleotides, including natural DNA and RNA as well as chemically modified XNA. The manuscript is clearly organized and well written, and will be a good read to the interested readers. I recommend publication in *Communications Chemistry* after a minor revision.

Response: We thank the reviewer for the very positive assessment of our manuscript.

1. Formatting issues, grammar errors and typos should be corrected.

a) Page 2, line 75, “the potential of this is method is showcased by ...”. The extra “is” should be deleted.

Response: We thank the reviewer for noticing this mistake which we have now corrected accordingly.

b) Page 4, line 154, “nucleobase of the sugar”. The word “of” should be changed to “or”.

Response: We thank the reviewer for noticing this mistake which we have now corrected accordingly.

c) Page 8, line 352, “aptamers and DNA enzymes”. I believe in this section the authors meant to talk about XNA enzymes instead of DNA enzymes.

Response: We thank the reviewer and we have changed DNA enzymes to XNA enzymes.

d) Page 10, line 428, “approach relies on the of a trinucleotide codon”. I believe the word “use” is missing here.

Response: We thank the reviewer and have changed this accordingly.

2. Consistency. Page 9, figure 5. The triphosphate groups in panels a and b should be presented in a consistent format.

Response: We thank the reviewer for this comment. We have now changed the triphosphate groups accordingly.

3. Page 9, the title of the last section. Since the following paragraph mainly describes a template-directed oligonucleotide synthesis, the word “de novo” in the title is not accurate.

Response: We thank the reviewer for this comment. We have now removed the word “de novo” from this title.

4. In addition to TdT and PUP, other template-independent polymerases are also potentially useful for controlled enzymatic synthesis of oligonucleotides. For example, DNA polymerase theta mutants have been used for enzymatic synthesis of RNA (analogue) sequences (Nucleic Acids Res., 2018, 46, 6271), which should be cited.

Response: We thank the reviewer for this important comment. We have now inserted the following sentence:

Progress to further improve synthetic yields and reach yet longer sequences by this approach, will certainly involve exploring other template-independent polymerases such as polymerase theta.⁸⁷

We have also cited the reference as suggested.

87 Randrianjatovo-Gbalou, I. *et al.* Enzymatic synthesis of random sequences of RNA and RNA analogues by DNA polymerase theta mutants for the generation of aptamer libraries. *Nucleic Acids Res.* **46**, 6271-6284, doi:10.1093/nar/gky413 (2018).

Additional action :

We have revised Figure 1 with some minor edits.

REVIEWERS' COMMENTS:

Reviewer #1 (Remarks to the Author):

The authors of this comprehensive review have adequately answered my questions and addressed my comments, as well as that of the other reviewer. After revision, I encourage acceptance of this manuscript.